# Microstructure and Corrosion Behavior of Sn–Zn Alloys

**DOI:** 10.3390/ma15207210

**Published:** 2022-10-16

**Authors:** Žaneta Gerhátová, Paulína Babincová, Marián Drienovský, Matej Pašák, Ivona Černičková, Libor Ďuriška, Róbert Havlík, Marián Palcut

**Affiliations:** Institute of Materials Science, Faculty of Materials Science and Technology in Trnava, Slovak University of Technology in Bratislava, J. Bottu 25, 91724 Trnava, Slovakia

**Keywords:** corrosion, microstructure, Sn–Zn alloys, electrochemical potentiodynamic corrosion test

## Abstract

In the present work, the microstructure, phase constitution, and corrosion behavior of binary Sn–xZn alloys (x = 5, 9 and 15 wt.%) were investigated. The alloys were prepared by induction melting of Sn and Zn lumps in argon. After melting, the alloys were solidified to form cast cylinders. The Sn–9Zn alloy had a eutectic microstructure. The Sn–5Zn and Sn–15Zn alloys were composed of dendritic (Sn) or (Zn) and eutectic. The corrosion behavior of the Sn–Zn alloys was studied in aqueous HCl (1 wt.%) and NaCl (3.5 wt.%) solutions at room temperature. Corrosion potentials and corrosion rates in HCl were significantly higher compared to NaCl. The corrosion of the binary Sn–Zn alloys was found to take place by a galvanic mechanism. The chemical composition of the corrosion products formed on the Sn–Zn alloys changed with the Zn weight fraction. Alloys with a higher concentration of Zn (Sn–9Zn, Sn–15Zn) formed corrosion products rich in Zn. The Zn-rich corrosion products were prone to spallation. The corrosion rate in the HCl solution decreased with decreasing weight fraction of Zn. The Sn–5Zn alloy had the lowest corrosion rate. The corrosion resistance in HCl could be considerably improved by reducing the proportion of zinc in Sn–Zn alloys.

## 1. Introduction

Soldering is a metallurgical joining technique that uses a filler metal solder to join metallic substrates [1]. The solder is a crucial component of the electronic assembly that provides electrical, mechanical, and thermal continuity. Traditional solders were based on Sn–Pb eutectic and near eutectic compositions [1,2]. These lead-based alloys had several advantages, including low melting point and low surface tension, facilitating substrate wetting. However, environmental and health concerns about lead have limited their practical use [3]. Lead is a toxic metal that has various adverse health effects. Its accumulation in living organisms may lead to disorders of the nervous system and serious retardation of neurological and physical development. Lead is also known to negatively impact cognitive and behavioral development. Furthermore, it reduces hemoglobin production resulting in anemia and hypertension. To avoid these serious health issues, a European Union directive 2011/65/EU Restriction of the use of hazardous substances in electrical and electronic equipment (RoHS EEE) and its amendment 2015/863 have been launched [1,4,5]. The directive severely limits the use of toxic metals (Pb, Hg, and Cd) and other dangerous substances in electrical and electronic equipment.

In lead-free solder alloys, Pb is substituted by other metallic elements located in neighboring groups of the periodic table. Of these elements, Zn, In, Bi and Sb have been used [6]. Noble elements (Cu and Ag) are also utilized in small concentrations. Zn forms a eutectic with Sn at 8.9 wt.% Zn [7]. The eutectic alloy has a low melting point (199 °C) which is very close to that of the eutectic Sn–Pb solder (183 °C). Therefore, the eutectic and near-eutectic Sn–Zn alloys are regarded as some of the most promising lead-free solder alternatives. Hypereutectic Sn–Zn alloys with 15–30 wt.% Zn have been studied as perspective protective coatings of steel [8,9,10,11,12,13]. The alloys have favorable friction and wear resistance and excellent solderability. Furthermore, they are ductile and have low electrical contact resistance [8,9,10,11,12,13]. The Sn–Zn alloy coatings on steel are environmentally friendly and combine barrier properties with the sacrificial properties of zinc (cathodic protection). They are regarded as promising alternatives to previously used toxic cadmium coatings.

The Sn–Zn alloys have a reasonable cost and favorable mechanical properties [1]. However, the major drawbacks are their low wettability and limited corrosion resistance [6]. The corrosion behavior of Sn–Zn solder alloys in NaCl solution was investigated by Liu et al. [14,15,16]. The authors found that the susceptibility to pitting was associated with the degradation of the passive film. The pitting formation in the Sn–9Zn solder alloy was divided into three stages [15]. In the first stage, the Cl^−^ anions from the solution were adsorbed on the surface of the passive film. The anions subsequently penetrated the film through intergranular boundaries. In the second stage, tiny cracks on the surface of the passive film were formed and penetrated the film inward. In the final stage, a localized breakdown of the passive film occurred due to the increase in internal stress.

The Sn–Zn alloys are prone to oxidation since Zn is an electrochemically active element [17]. Zn is easily oxidized to ZnO and Zn(OH)_2_. A recent review indicated that the corrosion behavior of Sn–Zn alloys is dependent on the amount of Zn [18]. The pitting susceptibility increases with increasing Zn concentration. The inferior pitting corrosion resistance of Sn–Zn alloys is usually ascribed to an increased amount of coarse Zn-rich phases as more defects are formed along the intergranular boundaries between Zn-rich grains and the Sn matrix [15]. An increase in the Zn concentration in Sn–Zn alloys also results in the pore and crack formation in the corrosion products. Therefore, an inward transport of chlorine anions from the outer layer through the Zn-rich grains is possible, resulting in weak corrosion resistance. These negative effects can be mitigated by lowering the Zn content in the alloys [19]. If the amount of Zn-rich phase is minimal compared to the volume of Sn–matrix, Sn is allowed to dissolve. The corrosion layer becomes more homogeneous and denser for a higher Sn content. Therefore, the Sn–Zn alloys with higher Sn concentration are more attractive for soldering in terms of their higher corrosion resistance. Furthermore, hypoeutectic Sn–Zn alloys have the advantage of better wettability and reliability [20,21]. Therefore, the hypoeutectic Sn–Zn alloys are regarded as promising lead-free solder substitutes in harsh environments.

Most previous papers have studied the corrosion behavior of Sn–Zn alloys in NaCl solution [14,15,16,18]. The corrosion resistance in alkaline and acidic environments have been significantly less explored. The corrosion susceptibility of Sn–Zn solder alloys in alkaline KOH solution was studied in references [22,23]. The authors found that the corrosion potential of the alloys did not change significantly when the Zn content increased from 0 to 12 wt.% [22]. The corrosion rate, however, increased with increasing Zn concentration in the alloy. The corrosion products of SnO_2_, SnO and ZnO, were found on the surface of the Sn–Zn solder alloys after the polarization experiment, indicating a formation of a passive film [22,23]. The corrosion of Sn–9Zn alloy in nitric and sulfuric acids was studied by Mori et al. [24]. In the initial stage, a de-alloying of Zn was observed, and a porous Sn layer on the surface of the alloy was formed. In the next stage, both Zn and Sn corroded depending on their composition ratios. The corrosion behavior in HCl has been studied for the eutectic Sn–9Zn alloy only [25,26]. The authors observed a preferential Zn dissolution in the alloy. The passivation film was porous and consisted of several corrosion products. The film was insufficient to stop further corrosion attacks. To our best knowledge, the corrosion behavior of hypoeutetic and hypereutectic Sn–Zn alloys in acidic media has not been investigated yet.

Our goal is to find the corrosion resistance of a series of binary Sn–Zn alloys in a dilute HCl solution (1 wt.%). The results are compared with standard NaCl solution (3.5 wt.%). HCl is a strong reducing acid with multiple industrial applications [27]. The acid in a concentration of ~1% is also found in the gastric tract, where it helps to ease digestion and protect against infection [28,29]. The objective of our study is to investigate the effects of pH and chloride concentration on the corrosion behavior of Sn–Zn alloys. The aim of the present paper is to provide a fundamental understanding of the corrosion behavior of Sn–Zn alloys in acidic and saline solutions. Therefore, hypoeutectic, eutectic, and hypereutectic alloys have been studied. Such an approach has been used by several authors previously to study the wetting behavior and mechanical properties of Sn–Zn alloys [30,31,32]. The overall aim of our study is to investigate the effect of Zn concentration on corrosion behavior. It is hypothesized that decreasing the Zn weight fraction in the alloy will improve the corrosion resistance of Sn–Zn alloys in an HCl solution.

## 2. Materials and Methods

The Sn–5Zn, Sn–9Zn and Sn–15Zn alloys were prepared by melting Sn and Zn lumps (Camex, Měšice, Czech Republic) in pre-weighted concentrations. The melting was conducted in an induction furnace (Rajmont, Hradec Králové, Czech Republic) in an argon-protected atmosphere. Alumina crucibles (diameter 30 mm, Brisk, Tábor, Czech Republic) were used for sample mounting. After melting, the alloys were solidified to form cast cylinders. The as-cast alloys were subsequently ground with grades 600 and 1200 SiC papers to achieve flat surfaces. Furthermore, the alloys were polished with monocrystalline diamond suspension down to 1 μm surface roughness. The surface roughness of 1 μm was required by the standard for electrochemical corrosion testing to eliminate the negative effects of scratches and grooves induced by grinding [33]. The alloys’ microstructures were studied using a JEOL JSM-7600F scanning electron microscope (SEM, JEOL Ltd., Tokyo, Japan). A back-scattered electron mode (BSE) was used for imaging. The chemical composition of the alloys was studied with an Oxford Instruments (Bucks, UK) energy-dispersive X-ray spectrometer (EDS) using a Si(Li) detector X-Max 50 mm^2^ integrated into SEM. The spectrometer was operated by INCA software (Oxford Instruments NanoAnalysis, Bucks, UK). The hardness of the polished alloys was measured on a ZWICK 3212 testing machine (ZwickRoell, Brno, Czech Republic) with a load of 9.81 N and an indentation time of 15 s. Five measurements per alloy were made. Average hardness and standard deviations were calculated from individual measurements for each Sn–Zn alloy.

The phase constitution of the as-cast alloys was studied using a Philips PW 1830 X-ray diffractometer (XRD, Malvern Panalytical Ltd., Malvern, UK). The diffractometer worked with Bragg–Brentano geometry and used CoK_α1,2_ as a radiation source. Diffraction measurements were carried out in a 2-Theta range between 20° and 80°. A step size of 0.05° and a counting time of 98 s per step were used during diffraction experiments. The X-ray radiation was generated at 40 kV and 40 mA.

The polished alloy surfaces (surface area 530 mm^2^) were subjected to electrochemical corrosion testing at room temperature (21 ± 2 °C). The experiments were conducted in a 0.5 L double-walled vessel with a working electrode (Sn–Zn alloy), reference electrode and counter electrode, respectively. An Ag/AgCl electrode suspended in a saturated KCl solution served as a reference electrode. A platinum sheet (20 × 20 mm^2^) was used as a counter electrode. The corrosion experiments were carried out in aqueous HCl (1 wt.%) and NaCl (3.5 wt.%) solutions. The solutions were prepared by dissolving the respective weighted amounts of HCl and NaCl in deionized water. The solutions were not de-aerated before the experiment to simulate real environmental conditions the alloys may experience in service. The corrosion measurements were controlled with a PGU 10 V-1A-IMP-S potentiostat/galvanostat from Jaissle Electronic Ltd. (Waiblingen, Germany). An open circuit potential (OCP) of the alloys was recorded first, immediately after the sample immersion in the electrolyte. After stabilization, an electrochemical polarization was initiated using a sweeping rate of 1 mV s^−1^. The potential range of each polarization experiment was selected according to the OCP of the alloy. It covered a minimum of 500 mV around the OCP on both cathodic and anodic sides. The obtained polarization curves were analyzed by Tafel extrapolation [34]. By using this method, corrosion current densities and corrosion potentials of the alloys were obtained. Subsequently, the corrosion rates were calculated from the experimental corrosion current densities using Faraday’s electrolysis laws. After polarization, the microstructures and chemical compositions of the corroded alloy surfaces were studied using SEM and EDS.

## 3. Results and Discussion

### 3.1. Phase Constitution and Microstructure of the Alloys before Corrosion

The phase constitution of the Sn–xZn alloys was studied by room-temperature X-ray diffraction. The XRD patterns of the alloys are presented in Figure 1. The XRD patterns of pure metals (Sn and Zn) are included in the same figure for comparison. The XRD patterns of Sn–xZn alloys (x = 5, 9, 15 wt.%) were found to be composed of Sn and Zn. The peaks of Sn could be assigned to Sn(01-086-2265). The peaks of Zn were assigned to Zn(01-078-9363). No other phases have been identified in the XRD patterns. Based on this observation, it can be concluded that minor phases were below the detection limit of the XRD instrument and constituted less than a few percent of the volume of the alloys [35,36]. The intensity of Zn peaks increased with increasing Zn concentration in the alloy (Figure 1). The intensity of the peaks was directly proportional to the Zn weight fraction. As such, the peak intensity was not significantly influenced by preferential grain orientation, internal stresses or other factors [37].

The microstructure of the as-cast alloys was inspected by SEM. The results are presented in Figure 2. The images were acquired in a BSE mode to yield element resolution. The Sn–5Zn alloy was composed of a light-colored dendritic microconstituent and a eutectic (Figure 2b). The Sn–9Zn alloy had a eutectic microstructure (Figure 2c). The Sn–15Zn alloy was composed of a dark microstructure constituent surrounded by the eutectic (Figure 2d). The chemical composition of the constituents was studied by EDS. The results are presented in Table 1. The light-colored microconstituent of the Sn–5Zn alloy was, based on its chemical composition and the corresponding XRD pattern, assigned to the solid solution of Sn. The solid solution of Sn is abbreviated as (Sn) in this paper. The solubility of Zn in (Sn) was less than 0.1% and could not be detected via EDS. The dark microstructure constituent of the Sn–15Zn alloy was identified as a solid solution of Zn. The solid solubility of Sn in (Zn) was less than 0.1%. The eutectic found in the Sn–xZn alloys had a chemical composition of approximately 9.1% Zn and 90.9% Sn (Table 1).

The microstructures of metallic Sn and Zn are given in Figure 2a,e. The microstructure of Sn was homogeneous (Figure 2a). The material was free of impurities, and only Sn was found by EDS. The microstructure of Zn is presented in Figure 2e. The material was heterogeneous. At the grain boundaries of Zn, isolated particles were detected (Figure 2f). The volume concentration of the particles was relatively small (less than 2 vol.%). As such, they could not be detected by XRD (Figure 1). The chemical composition of the particles was studied by EDS. The results are presented in Table 1. The particles contained approximately 95.1 wt.% Zn and 3.9 wt.% Fe. Small amounts of other elements were also detected (0.7 wt.% P and 0.3 wt.% Ti). Based on the chemical composition, the particles were assigned to FeZn_13_. FeZn_13_ is an intermetallic phase commonly observed in Zn-Fe alloys [38]. FeZn_13_ forms in Zn alloys due to the minimal solid solubility of Fe in (Zn). Zinc is often recovered from the furnace dust from galvanized steel-making plants [39]. As such, Fe impurities are difficult to avoid completely.

### 3.2. Hardness Measurements

The hardness was measured on polished Sn–Zn alloy surfaces using the Vickers method [40]. The results are presented in Figure 3. An increase in hardness with an increasing amount of Zn was observed. Tin had the lowest hardness. With the addition of zinc, there was a gradual increase in hardness up to 12 HV1 for the eutectic alloy. This observation is in accordance with previously published studies [41,42]. The increase in hardness is caused by the addition of much harder Zn. Furthermore, a strengthening of the grain boundaries also takes place in the Sn–Zn alloys. After exceeding 9 wt.% Zn, the increase in hardness was not significant. This observation is related to the formation of a large, needle-like (Zn) phase present in the Sn–15Zn alloy (Figure 2d). It has been previously observed that the large (Zn) needles no longer contribute to alloy strengthening [30,43].

### 3.3. Corrosion Experiments

The corrosion behavior of the Sn–Zn alloys was studied by electrochemical methods. Corrosion experiments were carried out in aqueous HCl (1 wt.%) and NaCl (3.5 wt.%) solutions to study the effects of H^+^ and Cl^−^ concentrations on the electrochemical properties. The OCPs of the alloys measured in an aqueous NaCl solution are given in Figure 4a. The OCPs of metallic Zn and Sn were also measured and are included in the same figure for the sake of comparison. In NaCl, the OCPs of the Sn–Zn alloys were found to increase in the following order:Sn–15Zn ˂ Zn ˂ Sn–9Zn ˂ Sn–5Zn ˂˂ Sn(1)

The OCPs of the Sn–Zn alloys were comparable to those of Zn. The OCP of Sn was significantly higher compared to those of Zn and Sn–Zn alloys. This observation reflects the higher standard electrode potential of Sn compared to Zn (E^0^ (Sn/Sn^2+^) = −0.13 V, E^0^ (Zn/Zn^2+^) = −0.76 V [17]).

The OCPs of the alloys in the HCl solution is presented in Figure 4b. The OCPs of pure metals are included for comparison. In HCl, the OCPs of the alloys were found to increase with increasing Sn concentration in the alloy:Zn ˂ Sn–15Zn ˂ Sn–9Zn ˂ Sn–5Zn ˂˂ Sn(2)

The OCPs of the Sn–Zn alloys were found to be close to those of Zn. The difference between the binary Sn–Zn alloys was relatively small. This behavior is in accordance with observations in the NaCl solution (Figure 4a) and suggests preferential corrosion (dissolution) of Zn in the alloys. Metallic tin is more noble than zinc. The OCP of Sn was significantly higher compared to the remainder of the materials (Figure 4b).

After the OCP measurement, the alloys were subjected to potentiodynamic polarization. Potentiodynamic polarization is an electrochemical method where the progress of the reaction is monitored by a potentiostat. The method provides a number of parameters and provides useful information about the reaction mechanism [33]. In the polarization experiment, three electrodes are positioned in a glass vessel cell. The electrodes include a working electrode (studied alloy), a counter electrode (inert Pt foil) and a reference electrode (saturated Ag/AgCl electrode). The voltage between the working electrode and reference electrode was systematically varied during the experiment (potentiodynamic polarization). The current running though the electrical circuit was measured by an inert counter electrode. The potentiodynamic polarization curves of the Sn–Zn alloys measured in NaCl and HCl solutions are presented in Figure 5. Pure elements (Sn, Zn) are also included. Each curve has two distinct parts that are separated by a well-defined corrosion minimum. The first region is the immune region; it is observed at potentials smaller than corrosion potential. In the immune region, the metal is thermodynamically stable [34]. The immune region is also called the cathodic region since cathodic reactions prevail at the metal surface. The second region, observed once the corrosion potential has been reached, is called the active region. In the active region, the metal actively corrodes. The active region is also called the anodic region since an anodic dissolution of the studied metal takes place in the studied solution. The metals can also passivate [44,45]. As such, a passivation region can also be found on the polarization curve. The passivation is pronounced by rapid current density decrease or stabilization observed at high potentials on the polarization curve [44,45].

The experimental polarization curves have been analyzed by Tafel extrapolation of the cathodic and anodic curves [34]. The corrosion parameters (E_corr_, j_corr_) obtained by Tafel extrapolation are presented in Figure 6 and Table 2. The experimental electrochemical parameters of the Sn–Zn alloys obtained by previous authors are also included in Figure 6 [15,19,46]. The corrosion potentials of the alloys in HCl increased with increasing Sn concentration. Furthermore, the corrosion currents of the Sn–Zn alloys decreased with increasing Sn concentration, indicating that a higher Sn content contributes to the higher corrosion resistance of the alloys.

The experimental corrosion current densities have been used to estimate the corrosion rates of the alloys. According to the Faraday law of electrolysis, the mass of the corroded metal is proportional to the current passing through the electric circuit [47]. As such, the following equation is valid
(3)$$m=\frac{E_{w}}{F}\;I_{\mathrm{corr}}t.$$

In Equation (3), *m* is the corroded mass, *I*_corr_ is the corrosion current, *t* is the corrosion time, *F* is a Faraday constant (96,485 C mol^−1^), and *E_w_* is an equivalent weight of the corroded metal/alloy. The equivalent weight of an alloy is defined as [48]
(4)$$\;E_{w}=\frac{1}{\sum \frac{z_{i}f_{i}}{A_{i}}}$$

In Equation (4), *z_i_*, *fi*, and A_i_ are the valences, weight fractions, and atomic masses of the constituent elements, respectively. The corrosion rate, *CR*, is defined as the thickness of the material, *x*, that corrodes over time. Therefore, it can be written as
(5)$$CR=\frac{x}{t}.$$

Taking into account the dimensions of the material, the corrosion rate becomes
(6)$$CR=\frac{x}{t}=\frac{V}{St}=\frac{m}{\rho St}.$$

In Equation (6), *V* is the volume corroded, *S* is the surface area, and *ρ* is the alloy density. The alloy density is defined as
(7)$$\rho =\sum f_{i}\rho_{i}.$$

In Equation (7), *f_i_* and *ρ_i_* are the weight fractions and densities of the constituent elements, respectively. By substituting Equation (3) with Equation (6), the corrosion rate becomes


(8)
$$CR=\frac{E_{w\;}j_{corr}}{\rho F}$$


In Equation (8), *j_corr_* is the experimental corrosion current density. Equation (8) has been used to estimate the corrosion rates of the alloys. The calculated CR values of the Sn–xZn alloys are given in Table 2. The CR varies between 0.2 and 279 mm/year. It increases with increasing Zn concentration in the alloy and decreasing the pH of the solution. The acidic solution is more aggressive compared to NaCl. Furthermore, Zn alloying increases the corrosion susceptibility of the alloys since Zn is less noble than Sn.

In the NaCl solution, the Sn–15Zn alloy was found to have the lowest corrosion rate. In the HCl environment, however, the Sn–5Zn alloy had the smallest corrosion rate (Table 2). The eutectic Sn–9Zn alloy was found to have worse corrosion resistance in both environments (Table 2). Although the eutectic Sn–9Zn alloy is suitable for soldering, one must consider its inferior corrosion resistance. The corrosion resistance of Sn–Zn alloys in HCl solution decreased with decreasing the weight fraction of zinc in the alloy. The hypoeutectic Sn–5Zn was more corrosion-resistant in HCl compared to the Sn–9Zn alloy. Its corrosion rate was 2-times lower compared to the eutectic alloy. Therefore, the hypoeutectic Sn–5Zn alloy can be considered a promising lead-free solder alternative in harsh conditions, where frequent acid attacks might be expected (chemical factories, outdoor environments polluted by acid rains).

The corrosion potentials of the Sn–Zn alloys increase with increasing Sn concentration (Figure 6a). This observation indicates that increasing Sn concentration contributes to the ennoblement of the alloys. The corrosion potentials and OCPs of the Sn–Zn alloys in the HCl solution are comparable (Table 2), demonstrating that the OCP is a good indicator of the corrosion potential in this solution. However, the corrosion potentials in NaCl are significantly smaller compared to the OCPs of the alloys (Table 2). A closer inspection of the polarization curves of the alloys in the NaCl solution shows that the Sn–Zn alloys tend to passivate in this solution as there is an intermediate decrease of the current density at potentials of −1200 to −1050 mV vs. Ag/AgCl (Figure 5a). This transient decrease is followed by an abrupt increase of current density at potentials higher than ~−1050 mV vs. Ag/AgCl, indicating a breakdown of the passive film. The passivation is pronounced by rapid current density decrease or stabilization at potentials higher than passivation potential (E_p_) on the polarization curve. At potentials higher than E_p_ the current started to increase abruptly. The increase was probably a result of passive film breakdown. The passive film breakdown was initiated by aggressive Cl^−^ anions. It may have resulted in localized corrosion (pitting) on the alloy surfaces.

Regions of immunity, activity, passivity, and transpassivity are indicated by different colors in Figure 7 for the sake of clarity. The OCPs of the Sn–xZn alloys in NaCl solution have been found at approximately −1000 mV vs. Ag/AgCl (Table 2). These values are in the pitting corrosion region (Figure 7a). Therefore, it can be concluded that the Sn–xZn alloys were subject to pitting corrosion immediately after their immersion in the NaCl electrolyte.

The corrosion potentials of the Sn–Zn alloys in the HCl solution were significantly higher compared to NaCl (Table 2). This observation can be explained by the equilibrium E-pH diagrams of Zn and Sn [49]. Both elements are prone to corrosion in acidic environments. In neutral solutions (pH ~ 7), however, Sn tends to passivate and form water-insoluble corrosion products. Al-Hinai et al. constructed a theoretical E-pH diagram of the Zn–Sn–H_2_O system [49]. The diagram is presented in Figure 8. The E-pH diagram is an equilibrium diagram showing the electrode potential (E) between a metal and its various oxidized species as a function of pH. The E-pH diagrams are derived from thermodynamic considerations. As such, they can be used to predict which species is thermodynamically more stable at a given potential and pH. The theoretical E-pH diagram of the Zn–Sn–H_2_O system shows that at acidic pH, a water-soluble Sn^2+^ is stable, suggesting a rapid dissolution of the alloys. At the pH of the NaCl solution (pH~7), however, a water-insoluble zinc-tin oxide is predicated on being more thermodynamically stable [49]. This oxide may form a passive layer on the metallic substrate. Therefore, the corrosion rate was significantly lower in the NaCl solution compared to HCl.

### 3.4. Inspection of Alloy Surfaces after Corrosion

To further probe the differences in the corrosion behavior of the Sn–Zn alloys, the material surfaces were inspected by SEM after potentiodynamic polarization. The results are presented in Figure 9. In the NaCl solution, a heterogeneous layer of corrosion products was formed on the Sn surface. In the HCl solution, the corrosion products grown on the Sn surface were larger, indicating a more pronounced corrosion attack. The chemical composition of the reaction products formed in the HCl solution was studied by EDS. The results are presented in Table 3. In Sn, a high concentration of oxygen and chlorine was detected. Therefore, it can be assumed that either tin oxychloride or tin hydroxychloride have been formed as corrosion products of Sn.

Tin dissolution takes place according to the following reaction [50]
Sn → Sn^2+^ + 2e^−^(9)

Oxidation (9) is compensated by a reduction of oxygen dissolved in the solution. The reduction of dissolved oxygen can be given by the following reaction
O_2_ + H_2_O + 2e^−^ → 4OH^−^(10)

During reaction (10), hydroxide anions are produced in the solution. OH^−^ anions can react with Sn^2+^ and form tin hydroxide
Sn^2+^ + 2OH^−^ → Sn(OH)_2_(11)

Tin hydroxide may further react with Cl^−^ and form a complex tin oxyhydroxychloride [19,20]
3Sn(OH)_2_ + 2Cl^−^ → Sn_3_O(OH)_2_Cl_2_ + H_2_O + 2OH^−^(12)

The formation of Sn_3_O(OH)_2_Cl_2_ on the Sn substrate has previously been observed by different authors [51,52]. The formation of a scale rich in Sn and O was also found on the surface of the Sn–5Zn alloy (Table 3). However, in this alloy, the corrosion products did not completely cover the substrate. They were preferentially located on the eutectic microstructure constituent (Figure 9c,d). Sn dendrites were free of corrosion products. This observation suggests that microgalvanic couples have been formed between the dendrites and the eutectic in the Sn–5Zn alloy. The formation of galvanic couples is schematically shown in Figure 10. The eutectic contained Zn. The Sn dendrites were nobler compared to the eutectic because of the higher electrode potential of Sn compared to Zn. As such, the dendrites were cathodic with respect to the surrounding eutectic. The eutectic played the role of an anode and was preferentially attacked by corrosion. The formation of corrosion products is indicated by a gray line (Figure 10). The corrosion products were preferentially located on the eutectic.

The Sn dendrites were excavated after corrosion (Figure 10). Some dendrites eventually fractured due to mechanical stresses from the surrounding corrosion products. Therefore, fractures of the dendrites have been observed on the post-corroded surface (Figure 9c,d).

The morphology of the corrosion products formed in the eutectic Sn–9Zn alloy is shown in Figure 9e,f. The corrosion products covered the alloy surface and constituted a passive layer protecting the alloy against further corrosion. However, spallation of the corrosion products was observed locally on the alloy exposed to the HCl solution, revealing the underlying metallic substrate (Figure 9f). The underlying surface contained a large number of pits. The presence of pits on the surface of the Sn–9Zn alloy explains the increase in the corrosion current density at large electrode potentials (Figure 5). The corrosion products formed on the alloy surface subjected to the HCl solution were excessively growing at high electrode potentials (Figure 5b). As a result of rapid growth, the corrosion products contained defects. Therefore, they spalled off and left the underlying substrate prone to further corrosion.

Cl^−^ is a readily adsorbable anion [14,15]. It can easily penetrate inwards the metal/oxide interface via intergranular boundaries of the oxide/hydroxide scale. As such, it may promote a localized breakdown of the passive film formed on the alloy at low potentials (Figure 7). Chlorine anions adsorb onto the passive film and penetrate the scale through grain boundaries and other structural defects [53]. The grain boundary diffusion of Cl^−^ is several orders of magnitude higher compared to the bulk diffusion. Furthermore, the Sn and Zn oxides formed on the Sn–Zn alloys are n-type semiconductors [16]. As they have an excess of positive charge carriers in bulk, a positively charged interface might have been formed between the oxidized alloy and the corrosive solution. The attraction of negatively charged Cl^−^ to the film is promoted. Furthermore, Cl^−^ ions can be readily combined with Zn^2+^/Sn^2+^ as they form chloride complexes. The complexes are less strongly bonded to the film. Such behavior may eventually result in the formation of microcracks in the passive film. The cracks may originate at the surface of the film and extend to the film/alloy interface. The Cl^−^ ions may diffuse to the film/alloy interface, where they react with Zn from the substrate and form chloride complexes. The continuous growth of these complexes generates internal stresses at the interface due to volume expansion [14,15]. The volume expansion and pore formation in the scale may lead to a localized breakdown, followed by pit formation at the initial Zn sites.

A re-passivation process may occur concurrently at the breakdown sites. The re-passivation (“healing of the film”) is attributed to the transport of OH^−^ which is known as a pitting-inhibiting anion [54]. Therefore, the re-passivation is more likely to occur in basic and neutral solutions. The concentration of OH^−^ decreases with decreasing pH of the solution. The re-passivation of the passive film was probably seriously retarded in the HCl solution. Therefore, the pitting of alloy surfaces in 1% HCl solution was more dominant despite the lower Cl^-^ concentration compared to the NaCl solution (3.5 wt.%).

The surface of the hypereutectic Sn–15Zn alloy after corrosion in NaCl solution is shown in Figure 9g. In this alloy, the needle-like (Zn) has been preferentially dissolved, leaving large cavities in the corroded alloy. It is suggested that galvanic corrosion has also been taking place in this alloy. The galvanic mechanism of the Sn–15Zn alloy is schematically depicted in Figure 11. The Zn-rich areas played the role of a local anode. The eutectic played the role of the local cathode because of the higher Sn concentration. The Zn-rich needles were preferentially dissolved. The remaining surface was covered with a layer of corrosion products (Figure 9g). The scale had a high concentration of Zn and O (Table 3). The layer of corrosion products is indicated by the gray scale in Figure 11.

The chemical composition of the corrosion products formed on the Sn–Zn alloys changed with increasing Zn weight fraction (Table 3). Alloys with a higher concentration of Zn (Sn–9Zn, Sn–15Zn and Zn) formed corrosion products with a high Zn atomic fraction. Zinc is prone to anodic dissolution according to the following reaction [50]
Zn → Zn^2+^ + 2e^−^(13)

During reaction (13), Zn cations are released into the solution. The Zn^2+^ cations may react with OH^−^ and Cl^−^ and form complex hydroxychlorides according to reactions (14) and (15)
Zn^2+^ + OH^−^ + Cl^−^ → Zn(OH)Cl(14)
5Zn^2+^ + 8OH^−^ + 2Cl^−^ → Zn_5_(OH)_8_Cl_2_(15)

It has been observed that zinc-based corrosion products may not serve as an efficient corrosion protection barrier for Sn–Zn alloys in chloride-containing environments [55].

Zinc does not form protective corrosion products in acidic environments. It is prone to corrosion. The corrosion rate of zinc in 1% HCl was the highest of all materials studied (Table 2). The Fe-rich impurities in Zn may have further accelerated the corrosion attack. In zinc, a serious corrosion attack has been observed. The grain interior was severely attacked (Figure 9i). Isolated particles have been found and identified as possible FeZn_13_ intermetallic phases (Figure 2f). The particles were preferentially located at grain boundaries of Zn. Due to the higher galvanic potential of Fe, the FeZn_13_ particles may act as local cathodes [56]. The presence of local cathodes has accelerated the degradation of the surrounding Zn matrix. The solid solubility of Fe in Zn is negligible [38]. Therefore, even a minor concentration of Fe can lead to the formation of FeZn_13_ in the Zn matrix. Fe is often unavoidable, as it is usually present in recycled zinc. A refinement of second-phase particles could be an option for reducing the corrosion rate. Zn alloys with refined FeZn_13_ have a considerably lower corrosion rate compared to coarse-grained particles [57].

## 4. Conclusions

In the present work, the microstructure, phase constitution and corrosion behavior of binary Sn–xZn alloys (x = 5, 9 and 15 wt.%) have been investigated. The corrosion experiments were carried out in aqueous HCl (1 wt.%) and NaCl (3.5 wt.%) solutions to study the effects of pH and pCl on corrosion behavior.

 The Sn–xZn alloys were found to be composed of Sn and Zn. The intensity of Zn peaks increased with increasing Zn concentration in the alloy. The Sn–9Zn alloy had a eutectic microstructure. The Sn–5Zn and Sn–15Zn alloys were found to be composed of dendritic (Zn) or (Sn) and eutectic. The binary Sn–Zn alloys were prone to pitting corrosion in NaCl solution. There was an intermediate decrease in current density observed at potentials from −1200 to −1000 mV vs. Ag/AgCl, indicating a passivation behavior. This transient decrease was, however, later followed by an abrupt increase of current density at potentials greater than −950 mV vs. Ag/AgCl. In the NaCl solution, the Sn–xZn alloys were subject to pitting corrosion immediately after their immersion in the electrolyte. The corrosion resistance of the Sn–Zn alloys increased with increasing Sn concentration. The corrosion potentials and corrosion rates of the Sn–Zn alloys in HCl were significantly higher compared to NaCl. The corrosion of the binary Sn–Zn alloys was found to take place by a galvanic mechanism. The (Sn) dendrites in the Sn–5Zn alloys were nobler compared to the eutectic. The eutectic played the role of an anode and was preferentially attacked by corrosion. In the Sn–15Zn alloy, the needle-like (Zn) was preferentially dissolved, leaving large cavities in the corroded alloy. The eutectic played the role of the local cathode in the Sn–15Zn alloy because of the higher Sn concentration. The eutectic was covered with a layer of corrosion products. The chemical composition of the corrosion products formed on the Sn–Zn alloys changed with the Zn weight fraction. Sn and the Sn–5Zn alloy formed reaction products rich in Sn. Alloys with a higher concentration of Zn formed corrosion products with a high Zn atomic fraction. The corrosion products formed on the surface of the binary Sn–Zn alloys in the HCl solution were prone to spallation. As a result of rapid growth, the corrosion products contained defects. The corrosion products formed in the acidic solution spalled off and left the underlying substrate prone to further corrosion. In the NaCl solution, the Sn–15Zn alloy had the lowest corrosion rate. In the HCl environment, however, the Sn–5Zn alloy was found to have the lowest corrosion rate. The eutectic solder had worse corrosion resistance in both environments. Although the eutectic Sn–9Zn alloy is suitable for soldering, one must consider its worse corrosion resistance. The corrosion resistance of Sn–Zn alloys in acidic environments can be increased by reducing the weight of zinc in the alloy. The hypoeutectic Sn–5Zn was more corrosion-resistant in HCl compared to the Sn–9Zn alloy.

The study has contributed to the general understanding of the corrosion behavior of lead-free solder alloys in acidic environments. It has been shown that the corrosion resistance of the Sn–Zn alloys can be improved by decreasing the weight fraction of Zn in the alloy. This result might be useful for practical applications of Sn–Zn alloys. The Sn–Zn alloys are used either as lead-free solder alloys or protective coatings of steel. As such, they may be exposed to harsh conditions where frequent acid attacks are expected (chemical factories, atmosphere polluted with acid rains). The decrease of the Zn weight fraction in the alloy could be an option how to improve the corrosion resistance. Future efforts in corrosion studies of Sn–Zn alloys could be directed toward a fundamental understanding of metastable pitting formation and preferential zinc dissolution. These two phenomena significantly contribute to lowering corrosion resistance of Sn–Zn alloys in acidic and saline environments.

## Figures and Tables

**Figure 1 materials-15-07210-f001:**
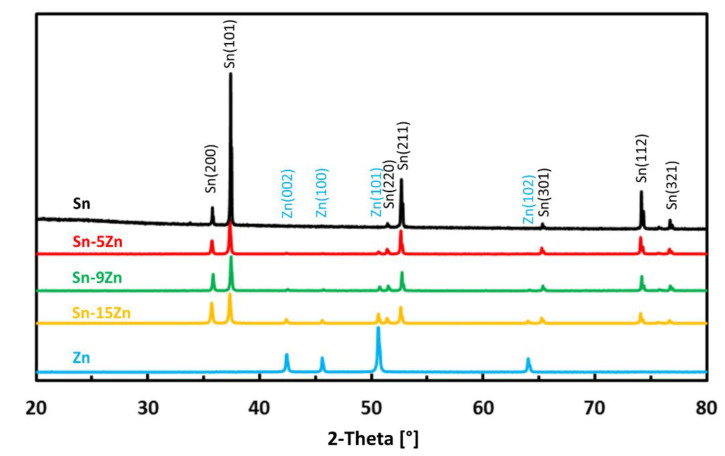
Room-temperature XRD patterns of as-cast Sn–Zn alloys.

**Figure 2 materials-15-07210-f002:**
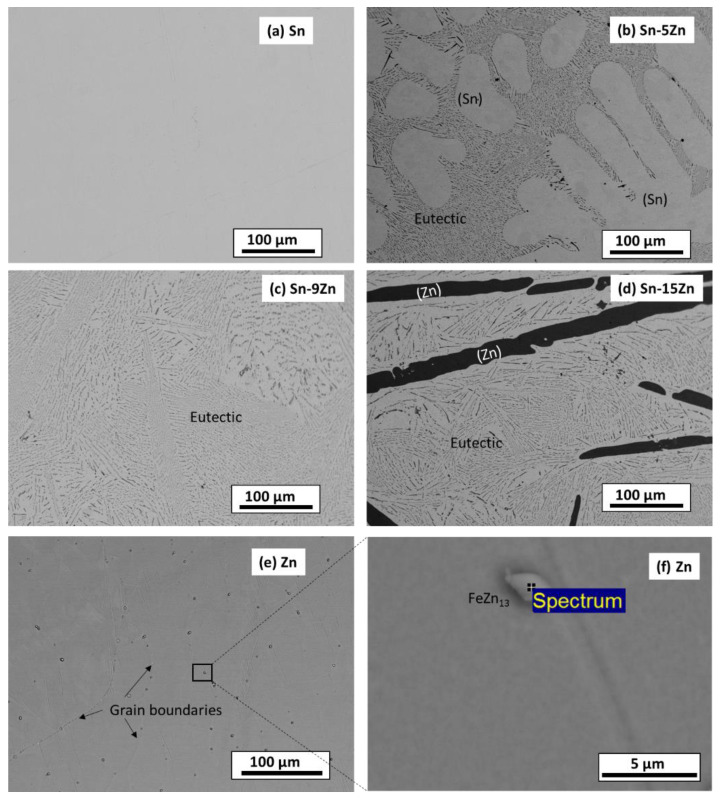
Microstructure of as-cast Sn–Zn alloys: Sn (**a**), Sn–5Zn (**b**), Sn–9Zn (**c**), Sn–15 Zn (**d**), and Zn (**e**,**f**).

**Figure 3 materials-15-07210-f003:**
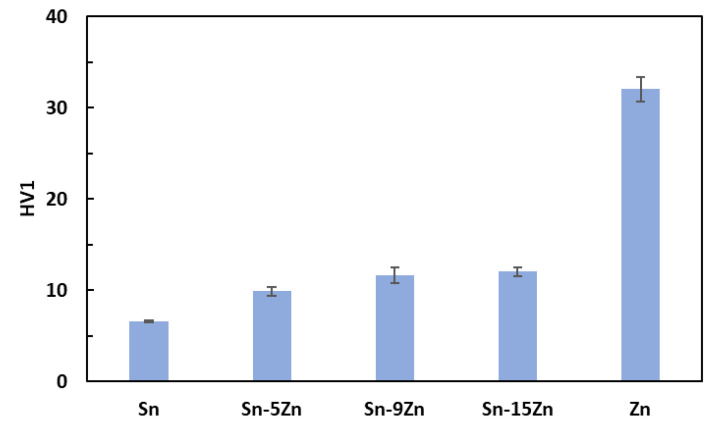
Vickers hardness of the Sn–xZn alloys.

**Figure 4 materials-15-07210-f004:**
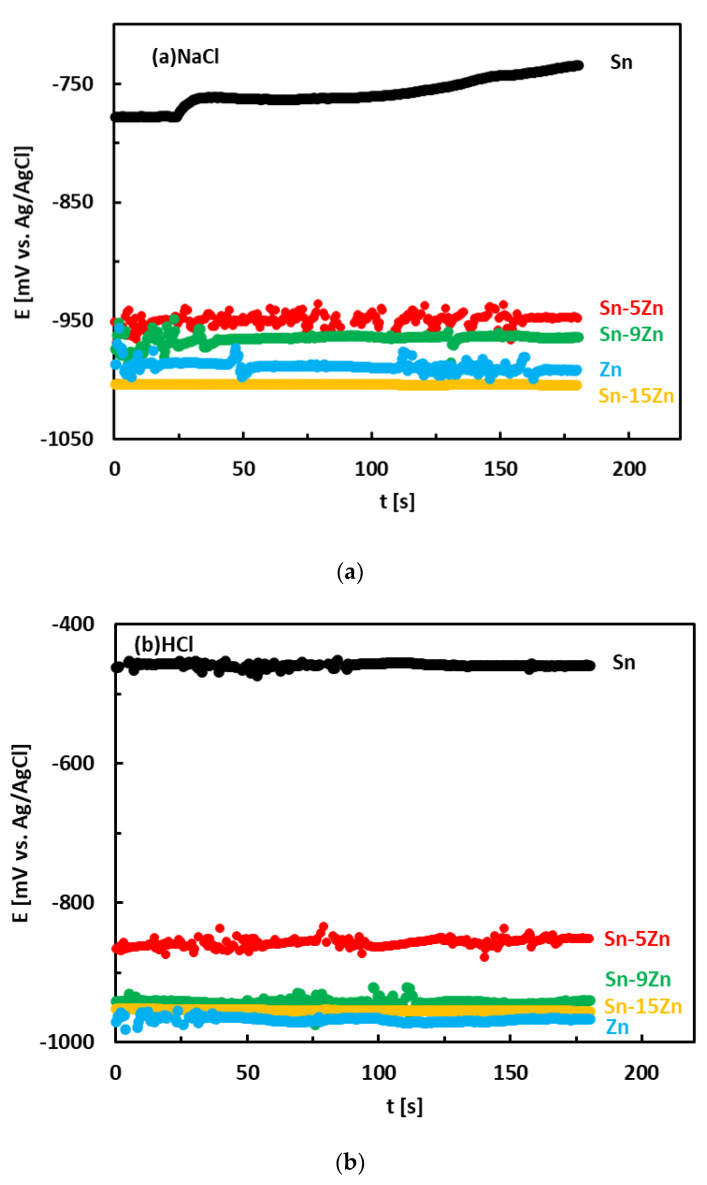
Room-temperature open circuit potentials of the Sn–Zn alloys in aqueous solutions of NaCl (**a**) and HCl (**b**).

**Figure 5 materials-15-07210-f005:**
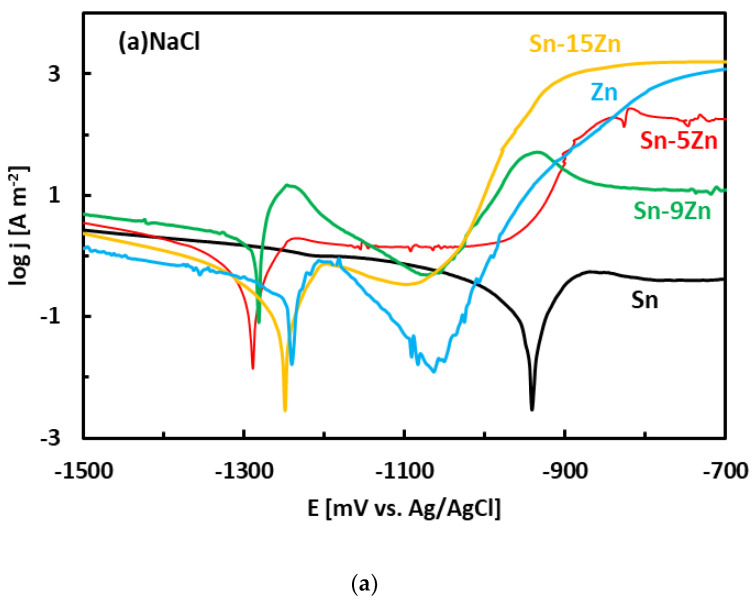
Potentiodynamic polarization curves of the Sn–Zn alloys in NaCl (**a**) and HCl (**b**) solutions.

**Figure 6 materials-15-07210-f006:**
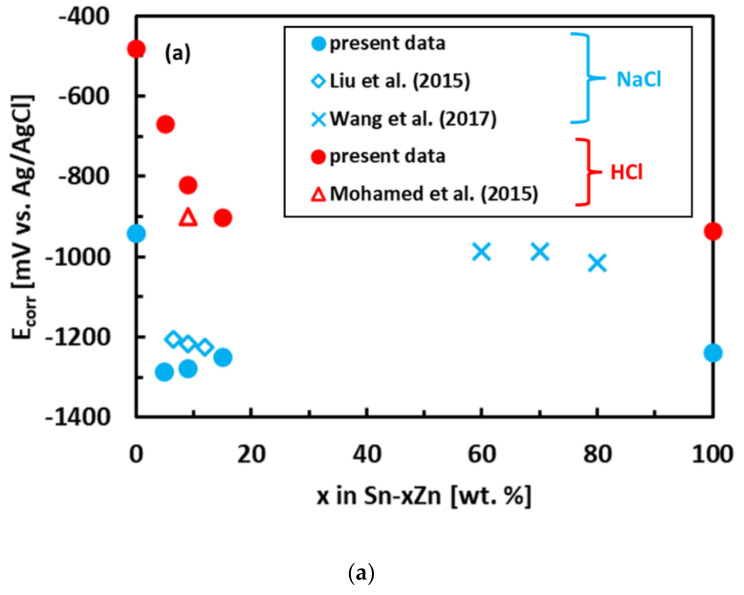
Corrosion parameters of Sn–Zn alloys: corrosion potential (**a**) and corrosion current density (**b**). Results of the present work are compared with ref. [15,19,46].

**Figure 7 materials-15-07210-f007:**
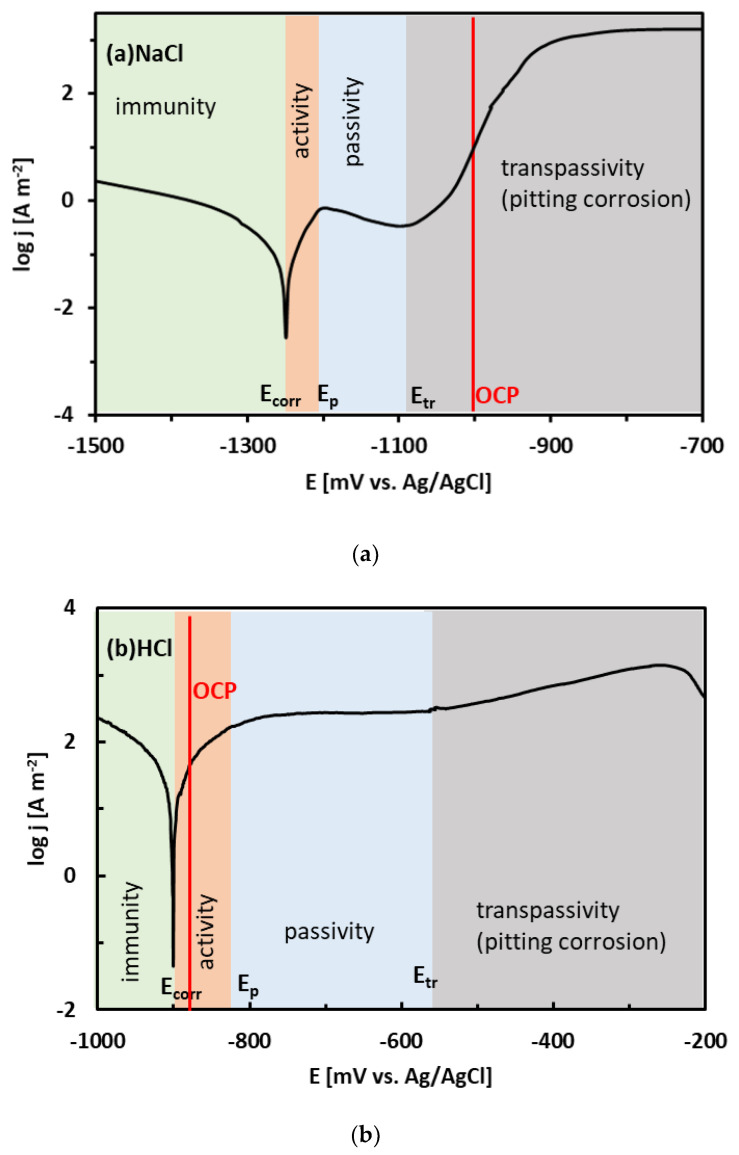
Position of the experimental open-circuit potential (OCP) of the Sn–15Zn alloy on the potentiodynamic polarization curve in NaCl (**a**) and HCl (**b**) solutions. Regions of immunity, activity, passivity, and transpassivity are indicated by different colors.

**Figure 8 materials-15-07210-f008:**
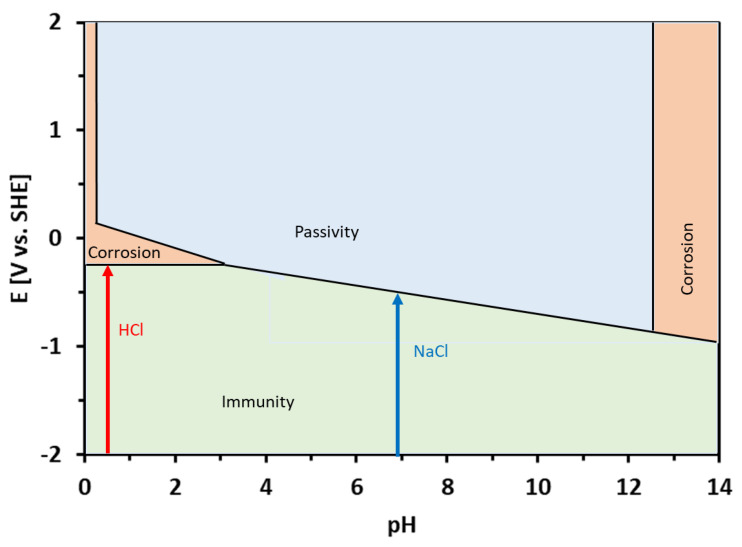
Theoretical equilibrium E-pH diagram of the Zn–Sn–H_2_O system, redrawn from ref. [49]. The pH values of the HCl and NaCl solutions are depicted by arrows.

**Figure 9 materials-15-07210-f009:**
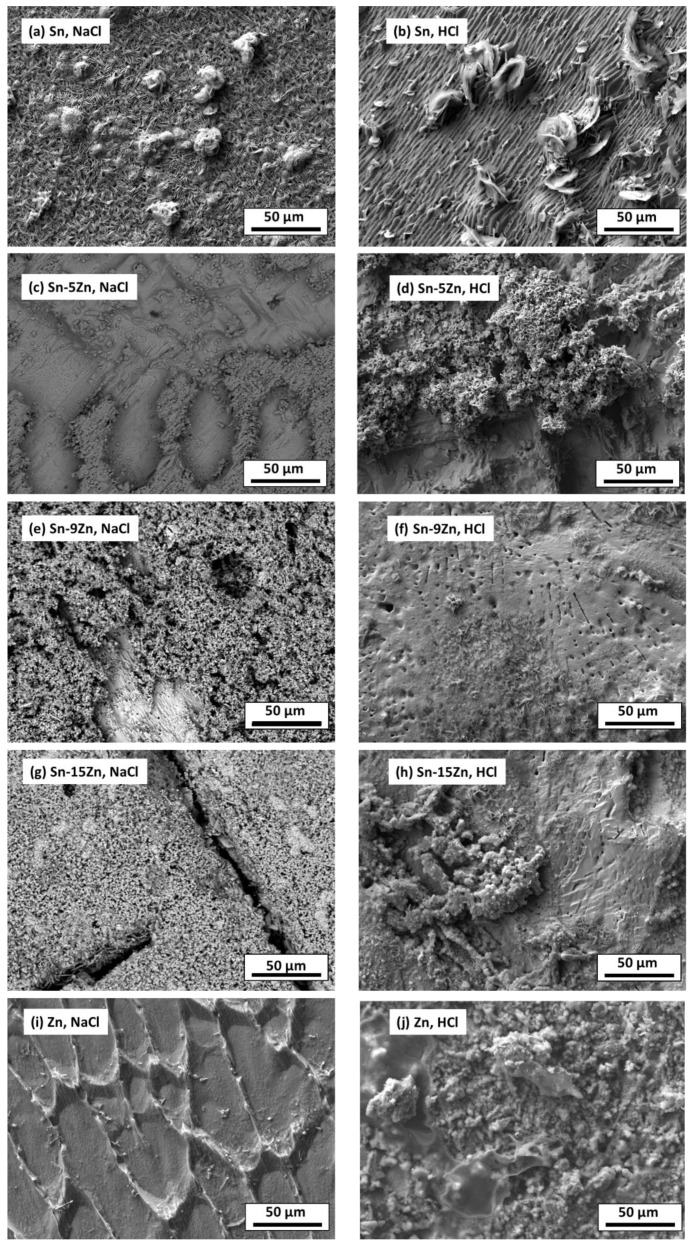
Microstructure of Sn–xZn alloys after corrosion in NaCl (**a**,**c**,**e**,**g**,**i**) and HCl solutions (**b**,**d**,**f**,**h**,**j**).

**Figure 10 materials-15-07210-f010:**
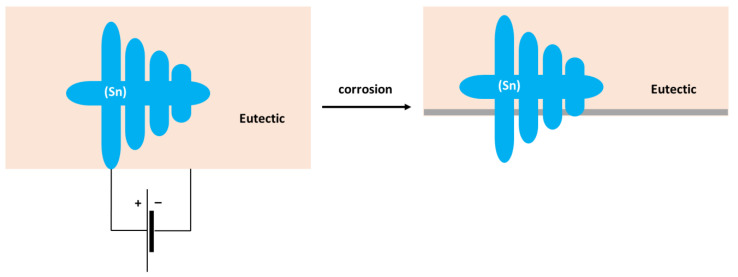
Schematic of microgalvanic cell formation in the Sn–5Zn alloy.

**Figure 11 materials-15-07210-f011:**
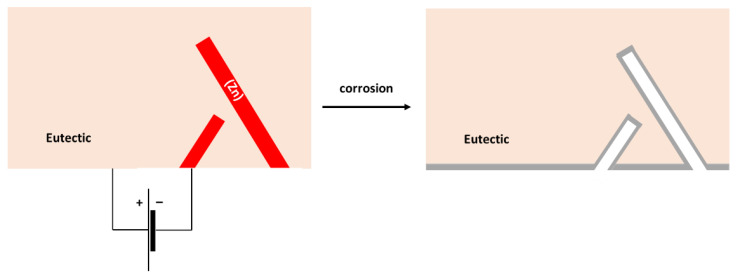
Schematic of microgalvanic cell formation in the Sn–15Zn alloy.

**Table 1 materials-15-07210-t001:** The chemical composition of the microstructure constituents (EDS point spectra).

Constituent	Chemical Composition [wt.%]				
	Sn	Zn	Fe	P	Ti
(Sn)	100.0	0.0	-	-	-
Eutectic	90.9	9.1	-	-	-
(Zn)	0.0	100.0	-	-	-
FeZn_13_	-	95.1	3.9	0.7	0.3

**Table 2 materials-15-07210-t002:** Physicochemical properties and experimental corrosion parameters of the Sn–xZn alloys studied in this work.

			3.5 wt.% NaCl			
Material	*E* _w_	*ρ*	OCP	*E* _corr_	*j* _corr_	*CR*
	[kg mol^−1^]	[kg m^−3^]	[mV vs. Ag/AgCl]	[mV vs. Ag/AgCl]	[A m^−2^]	[mm/year]
Sn	0.05935	7265	−735	−941	0.126	0.336
Sn–5Zn	0.05703	7259	−949	−1288	0.759	1.95
Sn–9Zn	0.05530	7254	−964	−1279	1.74	4.33
Sn–15Zn	0.05288	7246	−1004	−1249	0.100	0.239
Zn	0.03269	7140	−992	−1240	0.194	0.291
			**1 wt.% HCl**			
**Material**	** *E* _w_ **	** *ρ* **	**OCP**	** *E* _corr_ **	** *j* _corr_ **	** *CR* **
	**[kg mol^−1^]**	**[kg m^−3^]**	**[mV vs. Ag/AgCl]**	**[mV vs. Ag/AgCl]**	**[A m^−2^]**	**[mm/year]**
Sn	0.05935	7265	−460	−481	0.955	2.55
Sn–5Zn	0.05703	7259	−853	−668	6.03	15.5
Sn–9Zn	0.05530	7254	−940	−820	13.2	32.9
Sn–15Zn	0.05288	7246	−956	−900	33.9	80.8
Zn	0.03269	7140	−967	−935	186	279

**Table 3 materials-15-07210-t003:** Chemical composition of corrosion products formed on alloy surfaces in the HCl solution.

	Chemical Composition [at.%]			
Material	Sn	Zn	Cl	O
Sn	38.8 ± 13.2	-	14.9 ± 6.5	46.3 ± 7.3
Sn–5Zn	54.6 ± 14.7	2.6 ± 1.4	1.6 ± 0.9	41.4 ± 11.6
Sn–9Zn	10.8 ± 3.2	12.2 ± 2.9	4.7 ± 1.4	72.4 ± 1.1
Sn–15Zn	6.5 ± 4.0	16.7 ± 4.0	6.3 ± 1.7	70.5 ± 2.4
Zn	-	35.6 ± 6.0	4.5 ± 1.7	60.0 ± 5.1

## Data Availability

Data are available from the corresponding author upon request.

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
