# Peer review of "Microstructure and Corrosion Behavior of Sn–Zn Alloys"

_materials, 2022, doi:10.3390/ma15207210_

Round 1

Reviewer 1 Report

There is absolutely no justification for testing an alloy with a 15% zinc content, and the choice of an alloy with a 5% zinc content is also questionable, since the key point in using solder is its properties in the eutectic state.

Line 75 - why were the samples ground? Solders are not subjected to such treatment.

Why are research methods given in the conclusions (lines 371-376)?

The phrases "The Sn-xZn alloys were found to be composed of Sn and Zn. The intensity of Zn 377 signals increased with increasing Zn concentration in the alloy." (line 377) are puzzling: could it be otherwise? At the same time, the presence of iron, which affected the corrosion behavior, was not indicated.

Should the change in current density with potential (line 381-385) mean something? Both in the conclusions and in the very description of the results, dry data are given, and not what they mean for the study.

Are the corrosive products that cover the eutectic composition a potentially protective passive film, which as a result indicates their greatest corrosion resistance under dynamic conditions of long-term operation?

In general, there was no conclusion about which composition should be used according to the tests carried out.

Reviewer 2 Report

Review Reports

Manuscript title: Microstructure and Corrosion Behavior of Sn-Zn Alloys

Manuscript ID: materials-1936698

Review comments: The article is interesting to the readers. It describes about the the microstructure, phase constitution, and corrosion behavior of binary Sn – xZn alloys (x = 5, 9 and 15 wt.%). But it needs the following corrections before accepting its final publication.

1.    In abstract it needs to be re-written because it is mixed of past and present.

2.    In Introduction part it needs to add more related references to compare the present work to others.

3.    In Figure 3 insert the different color composition into the graph.

4.    In Figure 4 & figure 5 the same correction as figure 3.

5.    The English is need to edit moderately.

Reviewer 3 Report

-        Specify the supplier, where the materials were purchased.

-        Add a table of EDS test for alloys after obtain.

-        An important factor in the properties of an alloy is the mechanical characterization. Add some mechanical properties.

-        In the Introduction section, the authors cited the specific results of previous research and cited them adequately. However, they did not mention their shortcomings in previous research. In the Introduction section, the penultimate paragraph should contain common features of previous research. The shortcomings of previous research should also be pointed out, in general.

-        In the Introduction section, the last paragraph should contain the scientific contribution and scientific hypotheses of your research. Complete, further elaborate the scientific contribution and scientific hypotheses of your research. Be explicit. In addition to the goal of the research (which was written), the novelty in the context of the scientific contribution should be pointed out. Scientific contributions should be written based on the shortcomings of previous research in the literature. In this way, the authors will better emphasize novelty and scientific soundness.

-        Analyze and discuss possibilities of practical application.

-        In the conclusions, state the scientific contribution, the shortcomings of your methodology and future research.

Round 2

Reviewer 3 Report

Paper was improved.